# Zika virus infection in early pregnancy increases the likelihood of persistent neutralizing antibodies

Everton Falcão de Oliveira[1,2]*, Amanda Torrentes de Carvalho[3],
Fabio Antonio Venancio[2], Maria Eulina Quilião[4], Sanny Cerqueira de Oliveira Gabeira[5],
Silvia Helena dos Santos Leite[5], Margarida dos Santos Salú[5],
Sheila Maria Barbosa de Lima[6], Nathalia dos Santos Alves[7], Luma da Cruz Moura[7],
Waleska Dias Schwarcz[7], Adriana de Souza Azevedo[7], Luiz Henrique Ferraz Demarchi[8],
Marina Castilhos Souza Umaki Zardin[8], Gislene Garcia de Castro Lichs[8],
Deborah Ledesma Taira[8], Ana Isabel do Nascimento[2], Wagner de Souza Fernandes[9],
Natália Oliveira Alves[2], Aline Etelvina Casaril Arrua[9], Márcio José de Medeiros[10],
Rivaldo Venâncio da Cunha[2,11], Zilton Vasconcelos[5], Cláudia Du Bocage Santos-Pinto[1],
Karin Nielsen-Saines[12]*

1 Faculdade de Medicina, Universidade Federal de Mato Grosso do Sul, Campo Grande, Brazil,
2 Programa de Pós-Graduação em Doenças Infecciosas e Parasitárias, Universidade Federal de
Mato Grosso do Sul, Campo Grande, Brazil, 3 Laboratório de Imunobiologia das Doenças Infecciosas,
Departamento de Imunobiologia, Universidade Federal Fluminense, Niterói, Brazil, 4 Centro Especializado
em Reabilitação da Associação de Pais e Amigos dos Excepcionais, Campo Grande, Brazil, 5 Laboratório
de Alta Complexidade, Instituto Nacional de Saúde da Mulher, da Criança e do Adolescente Fernandes
Figueira, Fiocruz, Rio de Janeiro, Brazil, 6 Departamento de Desenvolvimento Experimental e Pré-Clínico,
Fiocruz, Rio de Janeiro, Brazil, 7 Laboratório de Análise Imunomolecular, Bio-Manguinhos, Fiocruz,
Rio de Janeiro, Brazil, 8 Laboratório Central de Saúde Pública de Mato Grosso do Sul, Secretaria de
Estado de Saúde de Mato Grosso do Sul, Campo Grande, Brazil, 9 Instituto de Biociências, Laboratório
de Parasitologia Humana, Universidade Federal de Mato Grosso do Sul, Campo Grande, Brazil,
10 Universidade Federal do Rio de Janeiro, Campus Macaé, Rio de Janeiro, RJ, Brazil, 11 Fundação
Oswaldo Cruz, Rio de Janeiro, Brazil, 12 David Geffen School of Medicine at the University of California,
Los Angeles (UCLA), Los Angeles, California, United States of America

* everton.falcao@ufms.br (EFdO); knielsen@mednet.ucla.edu (KN-S)

journal.pntd.0013906

BANGLADESH

**Peer Review History:** PLOS recognizes the
benefits of transparency in the peer review
process; therefore, we enable the publication
of all of the content of peer review and
author responses alongside final, published
articles. The editorial history of this article is
available here: https://doi.org/10.1371/journal.
pntd.0013906

## Abstract

### Background

Neutralizing antibodies (nAbs) play a central role in protection against Zika virus
(ZIKV) infection. Higher maternal ZIKV nAb titers during pregnancy have been asso-
ciated with reduced risk of congenital anomalies. However, limited data exist on the
long-term persistence of these antibodies in women infected during pregnancy and in
children exposed *in utero*.

### Methodology/principal findings

We conducted a cross-sectional serological study using stored serum samples from
a cohort of mother–child pairs with confirmed maternal ZIKV infection during preg-
nancy in Campo Grande, Brazil. ZIKV nAb titers were measured in samples collected

**Data availability statement:** All relevant data are included within the manuscript. The analyzed cohort is relatively small, and therefore, even de-identified data could potentially allow the identification of individual participants. For this reason, the dataset cannot be made publicly available. Access to the dataset may be granted upon justified request and subject to approval by the Research Ethics Committee of the Federal University of Mato Grosso do Sul (UFMS), which is the institutional body responsible for oversight of ethical use and protection of participant information. Third-party institutional contact: Research Ethics Committee – Federal University of Mato Grosso do Sul (UFMS) Cidade Universitária, Campo Grande – MS, Brazil, Email: cepconep.propp@ufms.br.

**Funding:** This study was financed in part by the Coordenação de Aperfeiçoamento de Pessoal de Nível Superior – Brazil (CAPES) (Finance Code 001) and Universidade Federal de Mato Grosso do Sul – Brazil (UFMS) to E.F.O., and the National Institute of Allergy and Infectious Diseases of the National Institutes of Health (NIH/ NIAID) through AI140718 to K.N.S. The funders had no role in study design, data collection and analysis, decision to publish, or preparation of the manuscript.

**Competing interests:** I have read the journal's policy and the authors of this manuscript have the following competing interests: EFO is an Academic Editor of PLOS One and PLOS Global Public Health. This does not alter our adherence to PLOS Neglected Tropical Diseases policies on sharing data and materials.

approximately 3–4 years after maternal infection using plaque reduction neutralization testing. Among 77 women, 66.2% (51/77) had ZIKV nAbs above the cutoff point, with higher titers observed in patients with first trimester infection. In contrast, only 2 of 72 children (2.8%) presented detectable ZIKV nAbs following clearance of maternal antibodies. No clear association was found between maternal nAb titers and adverse child outcomes.

## Conclusions/significance

Our findings suggest long-term persistence of neutralizing antibodies in most women infected with ZIKV during pregnancy, especially when infection occurred in the first trimester of pregnancy ZIKV-specific nAb persistence was rare in children with antenatal exposure once maternal antibodies waned, reinforcing concerns about limited postnatal protection in this group. These results underscore the importance of further studies to clarify the role of neutralizing antibodies in long-term protection and disease pathogenesis, particularly in endemic regions where the risk of reinfection or exposure to related arboviruses remains high.

## Author summary

Zika virus (ZIKV) infection during pregnancy can cause serious birth defects, and maternal neutralizing antibodies (nAbs) are thought to play a key role in protection. However, little is known about how long these antibodies last in mothers and their children after infection. In this study, we analyzed blood samples from women infected with ZIKV during pregnancy and their children, collected 3–4 years later. We found that most mothers still had detectable ZIKV nAbs, especially those infected in the first trimester. In contrast, very few children had these antibodies after maternal antibodies naturally disappeared, suggesting they may not have long-term protection against ZIKV. Our findings highlight the need for further research to understand how these antibodies affect protection and disease risks, particularly in regions where Zika and related viruses remain a threat. This work helps inform vaccination strategies and long-term care for children exposed to ZIKV before birth.

## Introduction

Several Zika virus (ZIKV) epidemics have been documented since the virus was first identified in 1947 in Uganda, Africa [1–3]. Attention to ZIKV was significant following the Latin American epidemic of 2014–2016, which led the World Health Organization to declare a Public Health Emergency of International Concern once the association between antenatal ZIKV exposure and microcephaly was described [4]. Together with other signs and symptoms, these findings formed the basis of what is now known as congenital Zika syndrome (CZS) [5].

Since the 2014–2016 epidemic in Latin America, much has been learned about the effects of antenatal ZIKV exposure. In children exposed during gestation, ZIKV may cause a spectrum of adverse outcomes ranging from severe teratogenic effects to milder neurodevelopmental sequelae [6]. Although the incidence of CZS declined substantially after 2017, the threat of future ZIKV epidemics remains, as many populations are still immunologically naive. Moreover, individuals who were children or adolescents during the 2015–2016 epidemic are now reaching reproductive age without prior exposure to the virus, thereby expanding the pool of susceptible individuals and potentially facilitating new outbreaks [3,7].

Importantly, the mean duration of protective Zika virus humoral immune responses is unknown [8], but there is some evidence suggesting that ZIKV infection may not confer long-lasting immunity in some adults and in children with antenatal exposure to the virus. This could render individuals susceptible to reinfection due to the absence of persistent neutralizing antibodies (nAb) [6,9,10]. This phenomenon also complicates the retrospective diagnosis of infection, as viral RNA can only be detected during the acute phase, and anti-ZIKV antibodies tend to wane over time and exhibit cross-reactivity with other arboviruses [6,11].

Neutralizing antibodies play a central role in protective immunity against ZIKV, preventing viral entry into host cells and contributing to viral clearance [9,10]. However, the magnitude and duration of these responses can vary substantially among individuals. Factors such as the timing of infection during pregnancy, the presence of cross-reactive antibodies to other flaviviruses, and the immaturity of the neonatal immune system may influence the persistence and effectiveness of these antibodies [9,10]. Understanding the dynamics of nAbs is therefore essential to assess long-term immunity, susceptibility to reinfection, and potential implications for vaccine design.

In 2018, following the Zika epidemic in Brazil, we initiated the recruitment of mother–child pairs with and without maternal and antenatal ZIKV exposure [12]. This cohort was followed until 2022, allowing for the characterization of clinical findings, management of seizures and epilepsy, and estimation of risks for both early and long-term outcomes associated with antenatal ZIKV exposure [12–14]. In these previous studies, maternal infection and prenatal ZIKV exposure were confirmed using secondary data routinely recorded by Brazil's epidemiological surveillance system. For the unexposed participants, in addition to secondary data, we applied a ZIKV infection screening algorithm to minimize the likelihood of erroneously classifying exposed participants as unexposed.

In the present study, we report results of antibody screening in women with confirmed gestational ZIKV during the epidemic and their children, using biological samples collected approximately three years after exposure. The aims of this study were twofold: (1) to assess the persistence of ZIKV nAb in women infected during pregnancy and in their children with antenatal ZIKV exposure; and (2) to explore whether maternal ZIKV nAb titers were associated with child health outcomes. We hypothesize that neutralizing antibodies against ZIKV in women tend to wane over time following infection, as observed in previous studies [9], but that this decline may occur slowly, with a baseline level of antibodies being maintained, as observed for other mosquito-borne flaviviruses, such as certain dengue virus (DENV) serotypes [15,16]. This sustained low-level immunity may explain the absence of Zika epidemics in Brazil since 2017 [6,7]. Potentially, the timing of infection during pregnancy may influence the durability of the neutralizing antibody response, as this factor has been frequently reported as a predictor of the occurrence of congenital Zika syndrome. For children, given the low frequency of anti-ZIKV antibody previously positivity reported [10], we hypothesize that long-lasting immunity against the virus does not occur, particularly when exposure occurred *in utero*.

## Methods

### Ethics statement

This study was approved by the Research Ethics Committee of the Federal University of Mato Grosso do Sul (CAAE: 26611819.2.0000.0021) under protocol number 6.329.609. Written informed consent to participate in the study was obtained from all participants, who were assigned alphanumeric codes used during all stages of the study to preserve data confidentiality.

## Study design

We studied neutralizing ZIKV antibodies in a subset of participants enrolled in a cohort study of mother-infant pairs with and without ZIKV in pregnancy whose children were evaluated over time for adverse pediatric outcomes, using a cross-sectional serological study design [14].

## Setting and participants

The study was conducted in Campo Grande, Mato Grosso do Sul state, Brazil, between 2018 and 2022. Details of the original cohort study were previously described [14]. Briefly, the initial study began in 2018 with the recruitment of children with laboratory-confirmed *in utero* exposure to ZIKV during the 2015–2018 period and their mothers matched to mother-infant pairs without ZIKV exposure. In the present analysis, only mothers infected with ZIKV during pregnancy and their children were included. As the study was initiated after the end of the Zika epidemic in Brazil, biological samples from the acute phase of maternal infection were not available for analysis. Blood collection occurred between October 2019 and June 2022. Due to the COVID-19 pandemic, study activities were suspended from February 2020 to August 2021. Blood samples were processed to obtain serum for antibody testing and plasma for viral RNA detection.

## Outcome measures, definitions, and data sources

For this study, outcomes were congenital abnormalities diagnosed at birth (early adverse outcome) and developmental delay in at least one of the following domains: cognitive, motor, and speech/language (long-term adverse outcome). The early adverse outcome was assessed based on live birth certificates, medical records, and a clinical evaluation performed by a child neurologist approximately 2–3 years after the child's birth to confirm retrospective findings. The long-term adverse outcome was assessed through a comprehensive evaluation of the child using the Brazilian neurodevelopmental tool Operationalized Portage Inventory [17,18], also performed by a child neurologist at age 3–5 years. Further details have been previously described [14]. Blood samples for the laboratory assays in this study were collected shortly after the clinical evaluation, between 2019 and 2021. Due to the COVID-19 pandemic, it was not possible to collect serial serum specimens except in the case of two mothers. For these two participants, only the result from the first collection was considered in the analysis.

## Laboratory analysis

All serum samples from mothers and children were tested for the presence of anti-ZIKV IgM and IgG antibodies through semi-quantitative serological analysis using the ZIKV Detect 2.0 Capture ELISA kit (InBios – Seattle, Washington, USA), following manufacturer's instructions. Serum samples with positive or indeterminate ELISA results for ZIKV antibodies underwent plaque reduction neutralization testing (PRNT). PRNTs were conducted on Vero CCL-81 (ATCC) cells in 24-well plates to detect neutralizing antibodies (nAbs). Titers were expressed as the reciprocal serum dilution capable of reducing 90% of plaque formation ($PRNT_{90}$). Samples with titers ≥ 1:140 were considered seropositive for ZIKV.

Quantitative real-time polymerase chain reaction (RT-qPCR) was performed on plasma samples to detect the presence of ZIKV RNA, using the Trioplex Real-Time RT-PCR Assay (Centers for Disease Control and Prevention, Atlanta, USA), according to the manufacturer's instructions, on the BD MAX System (Becton Dickinson, San Jose, USA). The following primers were used: ZIKVENV-F, sequence (5' to 3') GCT GGD GCD GAC ACH GGR ACT; ZIKVENV-R, sequence (5' to 3') RTC YAC YGC CAT YTG GRC TG; ZIKVENV-PROBE, sequence NEDCTGGAACAACAAAGAAGCAMGBNFQ.

We also included in the present analysis results for maternal neutralizing antibodies against yellow fever virus (YF nAb) since YF immunization is commonly administered in our setting. Antibody quantification was performed using the micro–plaque reduction neutralization test with horseradish peroxidase detection (µPRNT-HRP), following the protocol described by Reis et al. [19] and Simões et al. [20]. Neutralizing vaccine-induced antibody titers for YF were expressed as

the reciprocal serum dilution capable of reducing 50% of plaque formation. Samples with titers ≥1:100 were considered seropositive for YF, titers between 71 and 99 were classified as indeterminate (gray zone), and titers ≤70 were considered negative. For statistical analysis, titers were used both as a continuous variable and as a categorical variable (positive vs. indeterminate/negative).

All laboratory procedures are described in detail in Venancio et al. [14] and Oliveira et al. [21].

### Epidemiologic and clinical variables collected

Epidemiological and clinical-obstetric data which could impact both ZIKV nAb levels and infant outcomes were assessed. These variables included maternal age, prenatal care, maternal STORCH infections (syphilis, toxoplasmosis, rubella, cytomegalovirus, herpes, and HIV), gestational trimester of ZIKV infection, and maternal race/ethnicity. Data were obtained from maternal and child health records, including the prenatal and child health booklets (official documents completed by healthcare professionals during prenatal consultations and child follow-up visits) as well as from the Brazilian Live Birth Information System (SINASC). Information on birth outcomes and maternal complications was available in both sources, and our research team systematically cross-checked these data to ensure consistency. In cases of discrepancies, the information recorded in the maternal health booklet was prioritized, as it represents the primary source completed at the point of care.

In Brazil, maternal race (skin color) is self-reported and categorized into five groups: Black, Indigenous, Pardo (i.e., multiracial or mixed race), White, and Yellow. This classification was established in 1940 and has remained unchanged. Due to the small sample size in this study and considering the varying definitions of race/ethnicity across countries, we dichotomized this variable into White and Black/Mixed/Indigenous/Asian for the present analysis. In the Brazilian context, race and ethnicity serve as proxies for health and social inequalities.

### Statistical methods

Descriptive statistics were used to characterize the profile of study participants. Bivariate analyses were performed to compare the clinical and epidemiological characteristics of the mothers, as well as infant outcomes, according to the presence of ZIKV nAb. To describe and compare categorical variables between groups, relative frequencies were calculated, and the chi-square test or Fisher's exact test was applied when appropriate. For numerical variables, means, standard deviations, medians, and interquartile ranges (IQRs) were reported, and comparisons were made using the Mann–Whitney U test. Crude odds ratios (ORs) were also calculated to assess the strength and direction of associations.

A boxplot was used to illustrate ZIKV nAb titers according to the gestational trimester of maternal infection. A logarithmic scale was applied exclusively for graphical presentation. Spearman's correlation coefficient was used to assess the correlation between ZIKV and yellow fever nAb titers, both treated as continuous variables.

The analysis was performed using R software version 4.3.4 (https://www.r-project.org/), with the following packages: *tidyverse*, *descr*, and *epiR*.

### Results

In total, 77 mothers with laboratory confirmed ZIKV infection during pregnancy and 72 ZIKV-exposed children were eligible for assessment of ZIKV nAb assays, based on the availability of serum samples (Fig 1). Three pregnant women were asymptomatic and underwent ZIKV testing due to the detection of an adverse fetal outcome (i.e., congenital abnormality) during routine prenatal evaluations. None of these mothers had immunodeficiency, immunocompromised conditions, or autoimmune disorders at the time of recruitment into the study.

All mothers tested negative for ZIKV by RT-PCR during the post-infectious period. Most of them had non-reactive serologies for ZIKV: 57 (74%) were negative for anti-ZIKV IgM and 28 (34%) for anti-ZIKV IgG. One mother tested positive for anti-ZIKV IgM (1.3%), and 16 (21%) were positive for anti-ZIKV IgG (including the patient who was also IgM-positive).

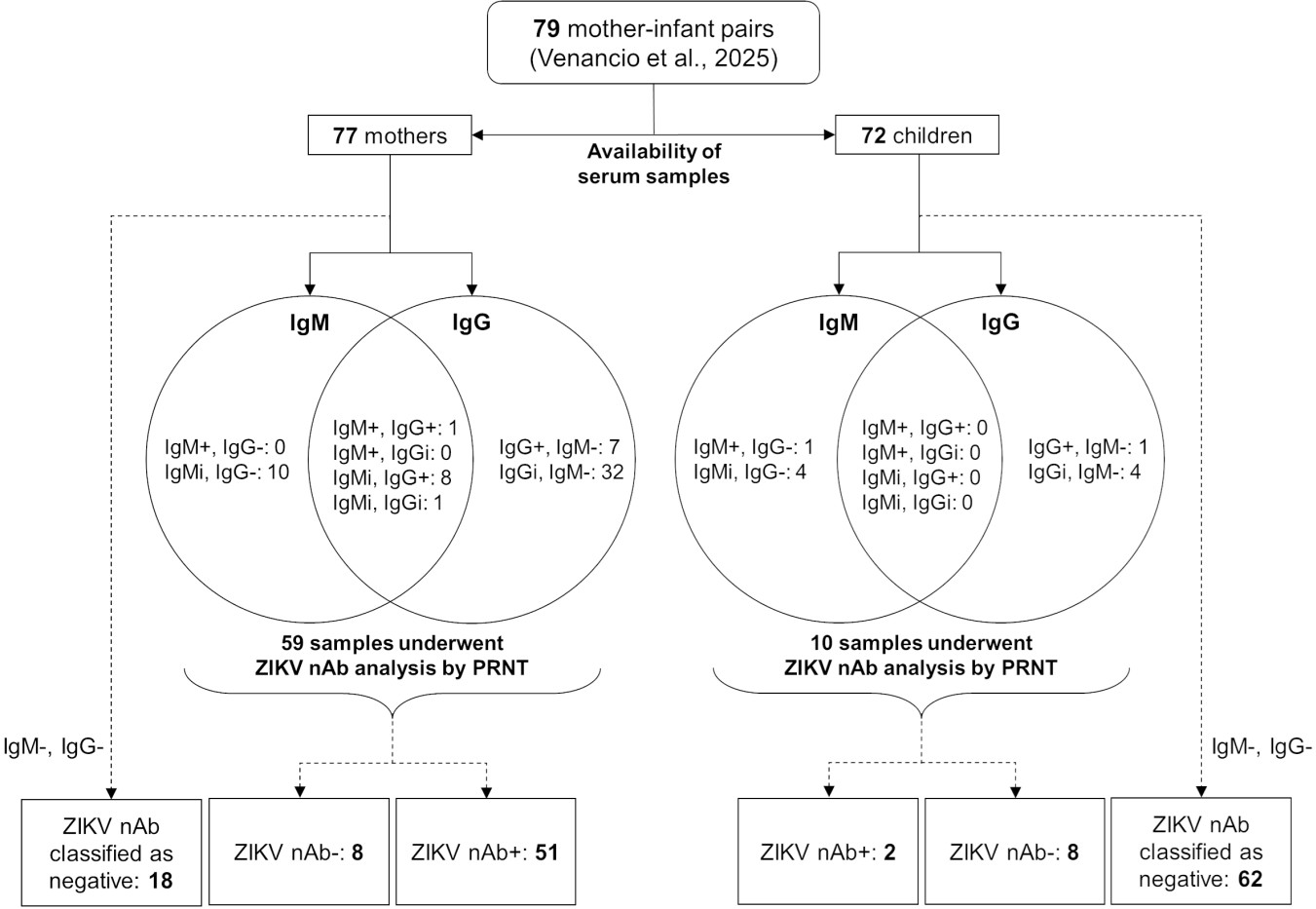

**Fig 1. Flowchart of participant recruitment and laboratory analysis.** IgM + : positive IgM result; IgM–: negative IgM result; IgMi: indeterminate IgM result; IgG + : positive IgG result; IgG–: negative IgG result; IgGi: indeterminate IgG result; ZIKV nAb: Zika virus neutralizing antibodies; PRNT: plaque reduction neutralization test.

Indeterminate results were observed in 19 mothers for IgM (25%) and 33 for IgG (43%). Based on these initial serological findings, PRNT was performed in specimens of 59 mothers (76.6%) who either had positive or indeterminate serologies for ZIKV. Among them, 51 (86.4% of those tested and 66.2% of the total cohort) had detectable ZIKV nAb on average 45.7 weeks after gestational infection. Among those who tested positive for ZIKV NAb, two had asymptomatic ZIKV infection during pregnancy. The mean ZIKV nAb titer was 1559.2 (SD: 4598.4) and the median was 532.5 (IQR: 283.2–897.0).

Clinical, obstetric, and epidemiological characteristics of mothers with ZIKV during pregnancy, according to the presence of ZIKV nAb, are summarized in Table 1. In this stage, women who tested negative for ZIKV serology and therefore did not undergo nAb testing were included in the 'No' category of ZIKV nAb titers. Among the variables analyzed, the gestational trimester at the time of infection was notably associated with the presence of ZIKV nAb after 45 weeks, with first-trimester infection increasing the likelihood of persistent immunity (OR: 4.72; 95% CI: 1.11–35.8) when compared to second-trimester infection. When comparing the first trimester to the combined group of second and third trimester infections, the odds of persistent immunity were even higher, with a slightly narrower confidence interval (OR: 5.10; 95% CI: 1.03–25.23). Asymptomatic infections (n = 3) were not included in this part of the analysis. Fig 2 shows ZIKV nAb titers according to the gestational trimester of infection.

**Table 1. Clinical, obstetric, and epidemiological characteristics of ZIKV-infected women during pregnancy and their offspring.**

| Variables | Total | ZIKV neutralizing antibodies | | OR (95% CI) | p-value* |
|---|---|---|---|---|---|
| | | Yes (*n* = 51) | No (*n* = 26) | | |
| Age during pregnancy (years) | | | | | |
| Mean (SD) | 28.2 (6.2) | 27.5 (5.8) | 29.7 (6.8) | 1.06 (0.98–1.15) | 0.167 |
| Median (IQR) | 28.0 (23.0–33.0) | 27.0 (23.0–31.5) | 29.5 (24.3–34.8) | | |
| Time since ZIKV infection (weeks) | | | | | |
| Mean (SD) | 44.6 (10.0) | 45.7 (7.5) | 42.2 (13.7) | 0.96 (0.92–1.02) | 0.248 |
| Median (IQR) | 43.0 (41.0–46.0) | 43.5 (41.0–46.0) | 43.0 (40.0–47.0) | | |
| Maternal race/ethnicity | | | | | 0.600 |
| White | 28 (36.4%) | 17 (33.3%) | 11 (42.3%) | Reference | |
| Black/Mixed/Indigenous | 49 (63.6%) | 34 (66.7%) | 15 (57.7%) | 0.68 (0.26–1.85) | |
| Gestational trimester of ZIKV infection** | | | | | 0.091 |
| First | 17 (23.0%) | 15 (30.6%) | 2 (8.00%) | Reference | |
| Second | 42 (56.8%) | 25 (51.0%) | 17 (68.0%) | 4.72 (1.11–35.8) | |
| Third | 15 (20.3%) | 9 (18.4%) | 6 (24.0%) | 4.57 (0.81–40.5) | |
| nAb Yellow Fever virus** | | | | | 0.789 |
| Yes | 46 (66.7%) | 29 (64.4%) | 17 (70.8%) | Reference | |
| No | 23 (33.3%) | 16 (35.6%) | 7 (29.2%) | 0.75 (0.24–2.19) | |
| Early outcomes in the offspring** | | | | | 1.000 |
| Yes | 12 (16.0%) | 8 (16.0%) | 4 (16.0%) | Reference | |
| No | 63 (84.0%) | 42 (84.0%) | 21 (84.0%) | 0.98 (0.27–4.20) | |
| Long-term outcomes in the children** | | | | | 0.933 |
| Yes | 29 (38.7%) | 20 (40.0%) | 9 (36.0%) | Reference | |
| No | 46 (61.3%) | 30 (60.0%) | 16 (64.0%) | 1.18 (0.44–3.31) | |

*Chi-Square Test or Fisher's Exact Test. **The total number of observations (N) varies across variables due to missing information. SD: standard deviation; IQR: interquartile range.

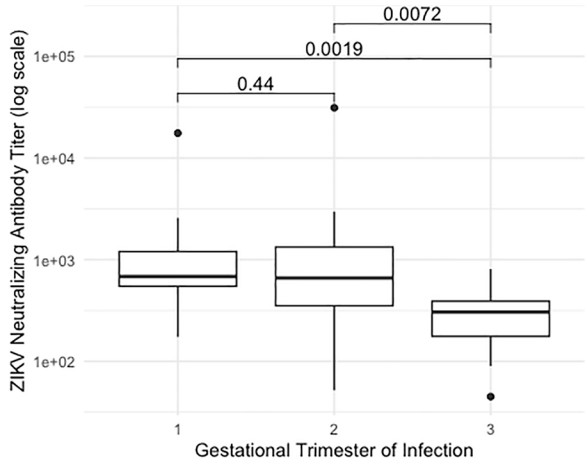

**Fig 2. ZIKV neutralizing antibody titers according to gestational trimester of ZIKV infection.**

Prior immunity to yellow fever was not associated with the presence of ZIKV neutralizing antibodies, nor were maternal race/ethnicity or age during pregnancy. Additionally, when analyzing the correlation between ZIKV nAb titers and YF nAb titers, no significant association was observed (rho = 0.23; p-value = 0.104).

The occurrence of congenital anomalies (early outcomes) and developmental delays (long-term outcomes) in the children of these women did not differ between mothers with and without neutralizing antibodies to ZIKV (Table 1). Clinical evaluations for the diagnosis of these outcomes were performed in 75 children, as two pregnancies ended in miscarriage.

Similar to what was observed in mothers, all children tested negative for ZIKV by RT-PCR in the post-infectious period at a median age of 40.0 months (mean: 40.0; SD: 11.0; IQR: 36.0–42.0). One child tested positive for anti-ZIKV IgM, and another for anti-ZIKV IgG. Eight children had indeterminate results for either ZIKV-specific IgM or IgG. Therefore, PRNT was performed using samples from 10 children. Among them, only two had ZIKV nAb titers at levels considered protective: one 61 month age-old child was IgG-positive and had congenital ZIKV syndrome (the mother of this child was not analyzed due to the unavailability of biological samples), and the other child had an indeterminate IgM result at 38 months of age (with no immediate or long-term outcomes reported; the mother had detectable ZIKV nAb). The child with a positive IgM result had a titer below the cutoff point for positivity. Antibody titers were markedly lower in children, with a mean ZIKV nAb titer of 140.0 (SD: 272.6) and a median of 10.0 (IQR: 10.0–47.0).

## Discussion

Neutralizing antibodies play a key role in protection against ZIKV, especially for pregnant women and their children, as congenital anomalies such as microcephaly and structural brain abnormalities were less frequent in infants born to mothers with higher ZIKV nAb titers during pregnancy [9]. However, there are still few published studies on the persistence and effectiveness of these antibodies over time in women infected during pregnancy and in children exposed *in utero*. Our study aimed to evaluate ZIKV nAb titers in serum samples collected approximately 3–4 years after maternal infection, and our findings suggest that the trimester of maternal infection may play an important role in the persistence of these titers.

More than half of the women in our cohort maintained high ZIKV nAb titers over time, suggesting the persistence of long-lasting immunity against ZIKV. Several studies have shown a robust maternal neutralizing antibody response developed during pregnancy [22–24], which remained detectable at least until birth and also in the post-delivery period [9]. Although there are conflicting findings regarding the association between maternal ZIKV nAb titers and the occurrence of CZS, as high maternal titers have been linked to the occurrence of microcephaly [23,24], it is important to consider the existence of other factors underlying both events (maternal ZIKV nAb and ZIKV-related birth defects), such as the trimester in which infection occurred [9,25,26] and prior exposure to other flaviviruses [27,28], which have been suggested as contributing factors, as observed in our study regarding the trimester of infection. Nonetheless, for serologies performed after birth, the association with infant outcomes may be less pronounced [9]. However, few studies have addressed this issue, highlighting the need for further investigation, especially at this timely moment, nearly 10 years after the onset of the Zika epidemic in Latin America.

Our findings regarding higher ZIKV nAb titers among mothers infected during early pregnancy should also be interpreted in light of the immunological adaptations that occur throughout gestation. These include maternal immune tolerance mechanisms and the general downregulation of the maternal immune response, particularly in the second and third trimesters, as well as ZIKV-associated maternal immune activation, especially during early pregnancy. Proteomic profiling has identified heightened inflammatory responses and a higher frequency of birth abnormalities in infants born to mothers infected in early pregnancy, suggesting that altered maternal immune responses may contribute to adverse child health outcomes [29].

The immunological implications of prenatal ZIKV exposure for a child's immune development during early life remain poorly understood [29]. While ZIKV neutralizing antibodies can persist for months or even years in adults, in children exposed *in utero*, nAb levels decline physiologically during the first months of life, reflecting the natural waning of passively

transferred maternal antibodies, even in those who exhibit an early IgM response. This suggests limited long-term protection and potential susceptibility to reinfection [10,30,31]. This context may help explain the low frequency of children with detectable ZIKV nAb in our study, which analyzed samples collected approximately three years after exposure.

Although one study has reported the persistence of ZIKV nAb in children exposed to ZIKV *in utero*, these findings may potentially be related to subsequent DENV infections after birth [31]. Our study did not assess the presence of anti-DENV IgM and IgG antibodies, and to our knowledge, none of the children included in our study were reported as suspected or confirmed dengue cases. However, Campo Grande is considered an endemic region for dengue, and we cannot rule out the possibility of DENV exposure in the two children who presented detectable ZIKV nAb in our study.

Our study has limitations, including the lack of evaluation for antibodies against other arboviruses that are endemic in the study region, such as dengue. Additionally, we did not have access to acute-phase samples from the mothers during infection, and therefore could not perform longitudinal analyses to assess ZIKV nAb titers over time. The relatively small sample size may also limit the generalizability of our findings. Future studies with larger cohorts and more comprehensive datasets, including information on additional maternal, perinatal, and postnatal factors, such as breastfeeding practices, are needed to strengthen the evidence on the persistence and effects of ZIKV nAbs.

Despite its limitations, this study offers valuable contributions to the understanding of long-term immunity following antenatal ZIKV exposure. By evaluating ZIKV neutralizing antibody titers approximately 3–4 years after maternal infection, our findings provide rare insight into the durability of humoral immune responses in a population affected by the Latin American Zika epidemic. The use of well-characterized clinical cohorts and biological samples collected years after exposure is a notable strength, particularly as global efforts to develop and evaluate ZIKV vaccines continue. Our data reinforce the hypothesis that maternal ZIKV nAb titers may be influenced by the timing of infection during pregnancy, which could have implications for maternal vaccine strategies aimed at maximizing fetal protection. These results underscore the importance of considering gestational timing and prior flavivirus exposure in future immunological and vaccine studies. As we approach a decade since the initial outbreak, our study highlights the need for continued surveillance, longitudinal immunologic assessments, and the identification of reliable immune correlates of protection to guide vaccine development and public health interventions in endemic regions.

## Conclusion

In conclusion, our findings indicate that a substantial proportion of women infected with ZIKV during pregnancy maintain detectable neutralizing antibody titers up to 3–4 years after infection, particularly those infected during the first trimester. These results suggest long-term persistence of maternal immunity, potentially influenced by the timing of infection. In contrast, ZIKV nAb titers were rarely detected in children exposed *in utero*, reinforcing evidence of limited long-term protection in this group. The lack of association between maternal nAb and child outcomes in our cohort highlights the need for further studies to elucidate the role of neutralizing antibodies in protection and pathogenesis, particularly in endemic settings, where the risk of subsequent ZIKV or other arboviral infections is higher.

## Acknowledgments

The authors are grateful to the team at the *Coordenadoria de Vigilância Epidemiológica* from the *Secretaria Municipal de Saúde de Campo Grande* for their assistance in locating and recruiting the study population. We also would like to acknowledge the children, their mothers and other family members who contributed so much to our understanding of Zika virus.

## Author contributions

**Conceptualization:** Everton Falcão de Oliveira, Fabio Antonio Venancio, Rivaldo Venâncio da Cunha, Zilton Vasconcelos, Karin Nielsen-Saines.

**Data curation:** Amanda Torrentes de Carvalho, Waleska Dias Schwarcz.

**Formal analysis:** Everton Falcão de Oliveira, Amanda Torrentes de Carvalho, Fabio Antonio Venancio, Maria Eulina Quilião, Sanny Cerqueira de Oliveira Gabeira, Silvia Helena dos Santos Leite, Margarida dos Santos Salú, Sheila Maria Barbosa de Lima, Nathalia dos Santos Alves, Luma da Cruz Moura, Adriana de Souza Azevedo, Luiz Henrique Ferraz Demarchi, Marina Castilhos Souza Umaki Zardin, Gislene Garcia de Castro Lichs, Deborah Ledesma Taira, Ana Isabel do Nascimento, Wagner de Souza Fernandes, Natália Oliveira Alves, Aline Etelvina Casaril Arrua, Márcio José de Medeiros, Cláudia Du Bocage Santos-Pinto, Karin Nielsen-Saines.

**Funding acquisition:** Everton Falcão de Oliveira, Karin Nielsen-Saines.

**Investigation:** Everton Falcão de Oliveira, Fabio Antonio Venancio, Maria Eulina Quilião, Sanny Cerqueira de Oliveira Gabeira, Silvia Helena dos Santos Leite, Margarida dos Santos Salú, Sheila Maria Barbosa de Lima, Nathalia dos Santos Alves, Luma da Cruz Moura, Waleska Dias Schwarcz, Adriana de Souza Azevedo, Luiz Henrique Ferraz Demarchi, Marina Castilhos Souza Umaki Zardin, Gislene Garcia de Castro Lichs, Deborah Ledesma Taira, Ana Isabel do Nascimento, Wagner de Souza Fernandes, Natália Oliveira Alves, Aline Etelvina Casaril Arrua, Márcio José de Medeiros, Zilton Vasconcelos, Cláudia Du Bocage Santos-Pinto.

**Methodology:** Everton Falcão de Oliveira, Rivaldo Venâncio da Cunha, Zilton Vasconcelos, Karin Nielsen-Saines.

**Project administration:** Everton Falcão de Oliveira, Cláudia Du Bocage Santos-Pinto.

**Resources:** Rivaldo Venâncio da Cunha, Zilton Vasconcelos, Karin Nielsen-Saines.

**Supervision:** Everton Falcão de Oliveira, Cláudia Du Bocage Santos-Pinto.

**Visualization:** Everton Falcão de Oliveira.

**Writing – original draft:** Everton Falcão de Oliveira, Karin Nielsen-Saines.

**Writing – review & editing:** Everton Falcão de Oliveira, Amanda Torrentes de Carvalho, Fabio Antonio Venancio, Maria Eulina Quilião, Sanny Cerqueira de Oliveira Gabeira, Silvia Helena dos Santos Leite, Margarida dos Santos Salú, Sheila Maria Barbosa de Lima, Nathalia dos Santos Alves, Luma da Cruz Moura, Waleska Dias Schwarcz, Adriana de Souza Azevedo, Luiz Henrique Ferraz Demarchi, Marina Castilhos Souza Umaki Zardin, Gislene Garcia de Castro Lichs, Deborah Ledesma Taira, Ana Isabel do Nascimento, Wagner de Souza Fernandes, Natália Oliveira Alves, Aline Etelvina Casaril Arrua, Márcio José de Medeiros, Rivaldo Venâncio da Cunha, Zilton Vasconcelos, Cláudia Du Bocage Santos-Pinto, Karin Nielsen-Saines.

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
