## [Decision Letter · Decision Letter 0]

30 Oct 2025

Response to Reviewers
Revised Manuscript with Track Changes
Manuscript

Kind regards,

Shaden Kamhawi

co-Editor-in-Chief

Paul Brindley

co-Editor-in-Chief

**Journal Requirements:**

At this stage, the following Authors/Authors require contributions: Everton Falcão de Oliveira, Amanda Torrentes de Carvalho, Fabio Antonio Venancio, Maria Eulina Quilião, Sanny Cerqueira de Oliveira Gabeira, Silvia Helena dos Santos Leite, Margarida dos Santos Salú, Sheila Maria Barbosa de Lima, Nathalia dos Santos Alves, Luma da Cruz Moura, Waleska Dias Schwarcz, Adriana de Souza Azevedo, Luiz Henrique Ferraz Demarchi, Marina Castilhos Souza Umaki Zardin, Gislene Garcia de Castro Lichs, Deborah Ledesma Taira, Ana Isabel do Nascimento, Wagner de Souza Fernandes, Natália Oliveira Alves, Aline Etelvina Casaril Arrua, Márcio José de Medeiros, Rivaldo Venâncio da Cunha, Zilton Vasconcelos, Cláudia Du Bocage Santos-Pinto, and Karin Nielsen-Saines. Please ensure that the full contributions of each author are acknowledged in the "Add/Edit/Remove Authors" section of our submission form.

2) We noticed that you used the phrase 'data not shown' in the manuscript. We do not allow these references, as the PLOS data access policy requires that all data be either published with the manuscript or made available in a publicly accessible database. Please amend the supplementary material to include the referenced data or remove the references.

- TM on page: 7.

4) Thank you for including an Ethics Statement for your study. Please state whether the consent obtained is verbal or written.

5) Please upload all main figures as separate Figure files in .tif or .eps format. For more information about how to convert and format your figure files please see our guidelines:

6) Thank you for stating "The dataset may be shared upon justified request to the authors, as it contains potentially sensitive information." For studies involving human research participant data or other sensitive data, we encourage authors to share de-identified or anonymized data. However, when data cannot be publicly shared for ethical reasons, we allow authors to make their data sets available upon request. For information on unacceptable data access restrictions, please see https://journals.plos.org/plosntds/s/data-availability#loc-unacceptable-data-access-restrictions.

7) Please amend your detailed Financial Disclosure statement. This is published with the article. It must therefore be completed in full sentences and contain the exact wording you wish to be published.

**Reviewers' comments:**

**Key Review Criteria Required for Acceptance?**

**Methods**

-Are the objectives of the study clearly articulated with a clear testable hypothesis stated?

-Is the study design appropriate to address the stated objectives?

-Is the population clearly described and appropriate for the hypothesis being tested?

-Is the sample size sufficient to ensure adequate power to address the hypothesis being tested?

-Were correct statistical analysis used to support conclusions?

-Are there concerns about ethical or regulatory requirements being met?

Reviewer #1: Everton Falcão de Oliveira and colleagues evaluated the persistence of ZIKV neutralizing antibodies during pregnancy, which were evaluated 3-4 years after infection. This study included evaluations of children and mothers.

The methodology consisted of collecting epidemiological data and data on the gestation period and birth of the children. The evaluation of ZIKV infection during gestation was carried out by the same team and included a series of assessments to determine the incidence of infection. The evaluation of nABs was performed at a single point in time between 3 and 4 years after birth. The population is well described and is appropriate for this type of study. The sample size is a limitation, but it is understandable for such epidemiological studies, particularly given the sample collection and the epidemiological information available. The statistical analysis was appropriate. and there are no concerns about compliance with ethical requirements.

Introduction:

Line 89: What do you mean when you say that new birth cohorts are reaching reproductive age and how this may affect future epidemics? I believe this happens every year before, during, and after the pandemic, and I don't think it's relevant to mention it.

Regarding the collection of epidemiological data on pregnancy and birth. Was any type of quality control carried out to guarantee the information? Who was responsible for receiving or storing it? To ensure the processing of the information, it would be important to provide these details.

Reviewer #2: The authors evaluated the persistence of neutralizing antibodies in antenatal ZIKV infections 3-4 years after infection. Their most important findings reveal that maternal antibody titers persist and are higher when infection occurred in the first trimester; data from infants are inconclusive. The study design and statistical analyses is according to the objective state.

Reviewer #3: First, I appreciate the opportunity to review an interesting and robust study.

My considerations will be brief and to the point, to contribute to the structural clarity of the text.

Method: In the "study design" section, keep only the description of the structure and type of research that was carried out and the method applied (it is cited in the abstract as a cross-sectional serological study).

I suggest moving the paragraph on lines 118-128 to the introduction, as well as the hypothesis, which should be implied in the section prior to the method.

Cite the reference for the statement: "This sustained low-level immunity could help explain the absence of Zika epidemics in Brazil since 2017" and include it in the introduction as well.

I suggest, if the authors agree, dividing the objective, since throughout the text association and persistence are evaluated separately. Dividing the objective into persistence of maternal nAbs and the association presented in the hypothesis.

**Results**

-Does the analysis presented match the analysis plan?

-Are the results clearly and completely presented?

-Are the figures (Tables, Images) of sufficient quality for clarity?

Reviewer #1: Results. he analyses are consistent with the plan presented.

Table 1. The N of the columns does not match the N of each estimated variable.

Although few children had ZIKV antibody titers, information on breastfeeding could be included, as it may influence the results.

The main limitations have already been described, primarily that the analysis is performed at a single point in time.

Reviewer #2: The analysis matches the plan designed, the results were presented clearly, and the figures summarize the main results.

Reviewer #3: The results are described clearly and objectively and meet the objectives set forth at the time of reading.

The boxplot is well-suited for use.

On line 295: would it be "first quarter" where it says just "quarter"? I believe this also occurred on line 305.

I would like a justification and explanation for the use of children's ages in months, we usually use weeks or months to account for gestational age and years after birth.

**Conclusions**

-Are the conclusions supported by the data presented?

-Are the limitations of analysis clearly described?

-Do the authors discuss how these data can be helpful to advance our understanding of the topic under study?

-Is public health relevance addressed?

Reviewer #1: The conclusions are well developed in accordance with the objectives set, but the limitations of the study are not mentioned.

Reviewer #2: The conclusions are supported by the data presented, and the limitations are clearly specified, such as not presenting an analysis of other arboviruses endemic to the study area and not presenting a longitudinal analysis of the neutralization assays. They demonstrate the importance of conducting an analysis of neutralizing antibody persistence for new antibody-based therapies and counteracting the effects of future epidemics.

Reviewer #3: The discussion and conclusion meet the objective and proposed hypothesis.

**Editorial and Data Presentation Modifications?**

Reviewer #1: My opinion on the editorial and data presentation modifications is “minor revisions,” as it is suggested that data on the breast milk of the children studied be included. Due to the absence of conclusive data on the children's neutralizing antibodies, the conclusions will not change, but the insertion of this data is relevant.

Reviewer #2: (No Response)

Reviewer #3: No further modifications.

**Summary and General Comments**

Reviewer #1: The study of the duration of ZIKV nABs after infection in utero, as well as their relationship to short- and long-term adverse effects in children, is relevant. The authors also argue the importance of such studies for vaccine development and preventing serious effects in future epidemics. Some of the limitations of the study are the sample size and the inclusion of some data that could improve the evidence presented here.

Reviewer #2: The ZIKV epidemic was a key moment in the search for alternatives to this virus in order to reduce or eradicate its effects on the neurodevelopment of children during pregnancy. At the time, there was nothing else to do but observe the effects and study the areas that should be explored in depth. Now there is evidence of long-term manifestations, but rather than knowing what happens, it is more important to know what to do to manage future epidemics. This study presents evidence of the persistence of neutralizing antibodies in mothers, which is relevant for therapies that can reduce the effects on children in future epidemics.

Reviewer #3: I suggest describing the role of neutralizing antibodies in more detail in the Introduction.

I suggest describing the specifics of the immune response in management in the Introduction so that it can be revisited in the discussion. This creates an interesting thread.

PLOS authors have the option to publish the peer review history of their article (what does this mean? ). If published, this will include your full peer review and any attached files.

**Do you want your identity to be public for this peer review?** For information about this choice, including consent withdrawal, please see our Privacy Policy .

Reviewer #1: No

Reviewer #2: No

Reviewer #3: No

**Figure resubmission:**

**Reproducibility:** To enhance the reproducibility of your results, we recommend that authors of applicable studies deposit laboratory protocols in protocols.io, where a protocol can be assigned its own identifier (DOI) such that it can be cited independently in the future. Additionally, PLOS ONE offers an option to publish peer-reviewed clinical study protocols. Read more information on sharing protocols at https://plos.org/protocols?utm_medium=editorial-email&utm_source=authorletters&utm_campaign=protocols

---

## [Decision Letter · Decision Letter 1]

29 Dec 2025

Dear Dr Everton Falcão de Oliveira,

We are pleased to inform you that your manuscript 'Zika Virus Infection in Early Pregnancy Increases the Likelihood of Persistent Neutralizing Antibodies' has been provisionally accepted for publication in PLOS Neglected Tropical Diseases.

Best regards,

Md. Kamrujjaman, Ph.D

Academic Editor

Michael Holbrook

Section Editor

Shaden Kamhawi

co-Editor-in-Chief

Paul Brindley

co-Editor-in-Chief

Reviewer's Responses to Questions

**Key Review Criteria Required for Acceptance?**

**Methods**

-Are the objectives of the study clearly articulated with a clear testable hypothesis stated?

-Is the study design appropriate to address the stated objectives?

-Is the population clearly described and appropriate for the hypothesis being tested?

-Is the sample size sufficient to ensure adequate power to address the hypothesis being tested?

-Were correct statistical analysis used to support conclusions?

-Are there concerns about ethical or regulatory requirements being met?

Reviewer #1: No further analysis

Reviewer #3: The objectives are well-defined and well-articulated with the study hypothesis. The study design, population, and variables meet the proposed goals.

**Results**

-Does the analysis presented match the analysis plan?

-Are the results clearly and completely presented?

-Are the figures (Tables, Images) of sufficient quality for clarity?

Reviewer #1: no further comments

Reviewer #3: The results are relevant for supporting research related to the role of neutralizing antibodies and the immune response against ZIKV.

**Conclusions**

-Are the conclusions supported by the data presented?

-Are the limitations of analysis clearly described?

-Do the authors discuss how these data can be helpful to advance our understanding of the topic under study?

-Is public health relevance addressed?

Reviewer #1: no further comments

Reviewer #3: The authors addressed all the required topics.

**Editorial and Data Presentation Modifications?**

Reviewer #1: Accept

Reviewer #3: No further modifications.

**Summary and General Comments**

Reviewer #1: no further comments

Reviewer #3: No further modifications.

PLOS authors have the option to publish the peer review history of their article (what does this mean? ). If published, this will include your full peer review and any attached files.

**Do you want your identity to be public for this peer review?** For information about this choice, including consent withdrawal, please see our Privacy Policy .

Reviewer #1: No

Reviewer #3: No

---

## [Editor Report · Acceptance letter]

Dear Dr. Nielsen-Saines,

We are delighted to inform you that your manuscript, "Zika Virus Infection in Early Pregnancy Increases the Likelihood of Persistent Neutralizing Antibodies," has been formally accepted for publication in PLOS Neglected Tropical Diseases.

Best regards,

Shaden Kamhawi

co-Editor-in-Chief

Paul Brindley

co-Editor-in-Chief
